# A Dynamic Prediction Framework for Urban Public Space Vitality: From Hypothesis to Algorithm and Verification

**Yue Liu [1] and Xiangmin Guo [2,\*]**

[1] Faculty of Sciences, Engineering and Technology, School of Architecture and Civil Engineering, The University of Adelaide, Adelaide, SA 5005, Australia; ylforapplication@163.com

[2] School of Architecture, Harbin Institute of Technology, Shenzhen 518055, China

\* Correspondence: xiangminguo@hit.edu.cn; Tel.: +86-13632572636

**Abstract:** Predicting and assessing the vitality of public urban spaces is crucial for effective urban design, aiming to prevent issues such as "ghost streets" and minimize resource wastage. However, existing assessment methods often lack temporal dynamics or heavily rely on historical big data, limiting their ability to accurately predict outcomes for unbuilt projects. To address these challenges, this study integrates previous methodologies with observations of crowd characteristics in public spaces. It introduces the crowd-frequency hypothesis and develops an algorithm to establish a time-dimensional urban vitality dynamic prediction model. Through a case study of the Rundle Mall neighborhood in Adelaide, Australia, the effectiveness of the prediction model was validated using on-site observation sampling and comparative verification. The prediction model framework allows for the determination of urban vitality within specific time ranges by directly inputting basic information, providing valuable support to urban planners and government officials during the design and decision-making processes. It offers a cost-effective approach to achieve sustainable urban vitality construction. Furthermore, machine learning techniques, specifically the decision tree model, were applied to case data to develop a set of preliminary algorithm tools, which enable output of reference urban vitality levels (high-medium-low).

**Keywords:** urban vitality; prediction model; algorithm; crowd-frequency

## 1. Research Background and Introduction

In recent decades, the emergence of "ghost streets" in various new towns, such as Forest City in Malaysia [1] and Lingang New Town in Shanghai, China [2], has raised concerns. These areas exhibit significantly lower usage rates than anticipated, leading to resource wastage [2,3]. Improving the vitality of such zones often necessitates costly redevelopment efforts, which may not always yield successful outcomes. Moreover, several well-established large cities worldwide, including those in the United States, China, Europe, and Japan, have experienced urban shrinkage issues over the past thirty years [4,5] and require revitalization efforts. Consequently, there is a pressing need to develop algorithms capable of accurately predicting and evaluating urban vitality based on design plans, thereby mitigating risks prior to actual construction and fostering sustainable development.

Defined as the "intensity of people's concentration" [6], urban vitality is widely regarded as a key objective in urban public space design [7]. It is commonly assessed by evaluating the built environment's capacity to facilitate activity or the density of people in a given space [8], with some adopting composite approaches [9]. However, at the meso-micro level of general research, urban vitality is predominantly defined by the presence of active individuals in public urban spaces [10]. Previous studies by Chen et al. [11], Lv et al. [12] and Guo et al. [13] have demonstrated that urban vitality exhibits periodic variations throughout different times of the day and varies between weekdays and

weekends. Li and Zhao (2023) further argued that the vitality of public spaces on weekdays and weekends respectively correlates with the built environment of urban spaces [14]. Given the heterogeneity of urban vitality across temporal and spatial dimensions, there is a need for a method capable of accurately assessing and predicting urban vitality across different time periods and spaces.

Existing methods for quantitatively assessing urban vitality (Figure 1) exhibit both strengths and weaknesses. On-site observational statistics methods [10,15] directly collect data but are cumbersome to implement and only evaluate temporal urban vitality at specific points. Conversely, methods based on statistical analysis of urban big data, such as mobile phone signal data [11], Tencent location information [16], and online map POI data [17], offer rich statistical results and accurately assess urban vitality in specific regions at particular times. However, these methods often suffer from lag, fragmentation, imbalance, and limitations due to factors such as privacy and ethics. Additionally, most big data-based evaluation methods fail to differentiate between individuals in urban public spaces and those indoors. For example, while streets in a residential area may appear deserted at night, many residents may still be indoors, resulting in ambiguity if such situations are classified as active. Moreover, these methods heavily rely on historical statistical data and lack the ability to predict future urban plans. Even when analyzing data from urban areas inhabited for less than five years, the results often fall short of reflecting the stable state [2].

Methods assessing urban public space vitality based on urban physical built environment information remain unaffected by project completion status and offer good predictive capabilities. For instance, Ye and Nes [18] proposed a method to calculate urban vitality based on urban morphology using Space Syntax and Space Matrix tools, and the Mixed-Use Index. Similarly, Guo et al. [19] developed a method to calculate urban vitality based on urban public space information, immediate buildings, and interaction ball (a tool for quantifying social interaction levels). However, these methods overlook the laws of human subjective behavior, resulting in oversights regarding behavioral differences caused by user characteristics and variations in human behaviors across different time dimensions. Consequently, these methods only provide static and general vitality predictions for a specific set of physical urban built environments, failing to deliver accurate spatial-temporal vitality results across different time periods [10].

Algorithmic methods, generated through machine learning from results of actual measurement methods using urban information, offer the advantage of simplicity. However, their usage requires inputting urban physical built environment information and POIs from online maps, which does not completely eliminate dependence on big data.

Therefore, there is a critical need to explore new research pathways for more accurate evaluation and prediction of urban spatial vitality across various time periods. This study draws upon the aforementioned research methods and introduces time variables to establish a new index system for evaluating urban vitality. By analyzing the behavioral characteristics and typical schedules of various user groups in urban spaces, the study comprehensively considers users' impact on urban vitality and establishes an analysis framework for crowd-frequency over time. The efficacy of the new method was validated using the Rundle Mall neighborhood in Adelaide, Australia, as a case study. This prediction model framework only requires input of physical built environment, characteristic of resident, and accessible situation to estimate the vitality situation of a specific urban public space at a given period. Additionally, to enhance the usability and facilitate better understanding of evaluation results, machine learning was conducted on case data using a decision tree model, resulting in an algorithm capable of predicting the reference vitality levels (high-medium-low), by inputting time and urban information.

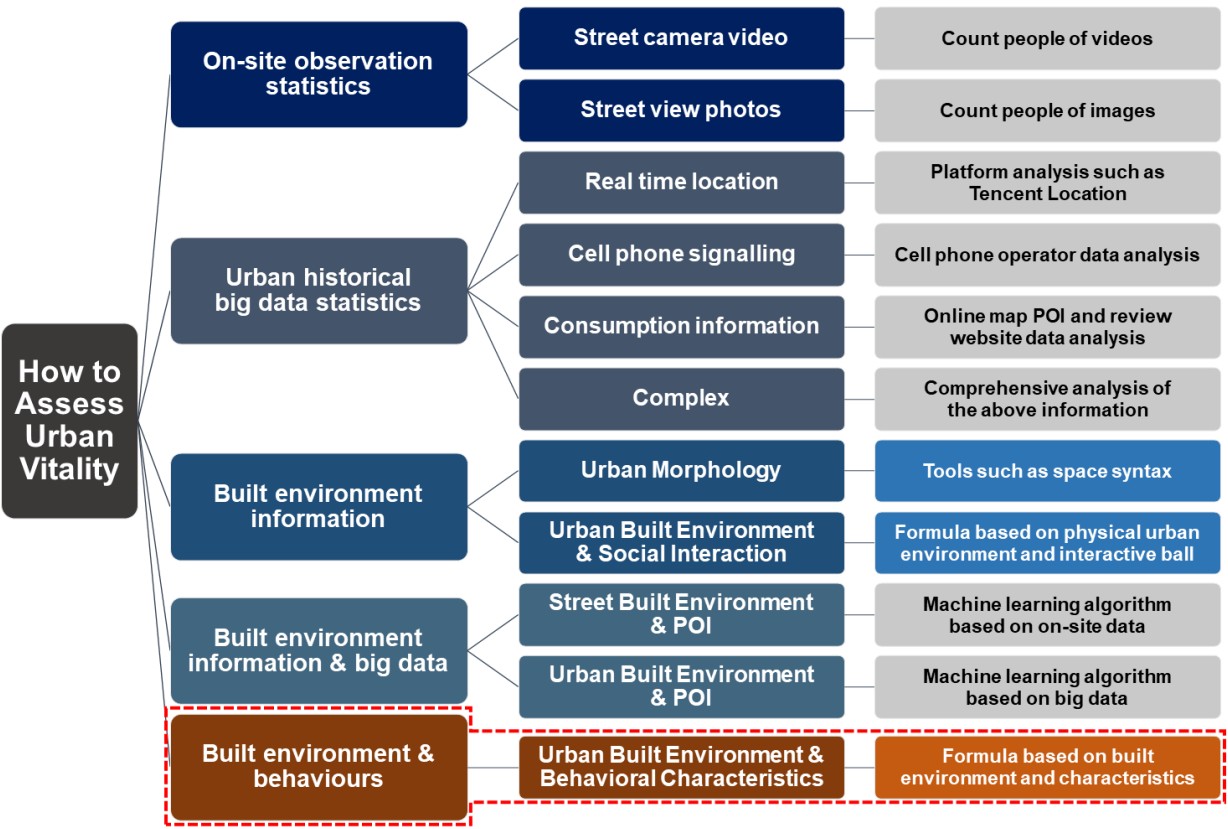

**Figure 1.** Review of urban vitality assessment methods and contribution of this study.

## 2. Theory of Time Dimension Dynamic Based on Crowd-Frequency

The existing methods for predicting spatial vitality are predominantly static and demonstrate limited accuracy in the temporal dimension. Although machine learning techniques can enhance accuracy by identifying patterns between real-time data and static prediction values across numerous cases, the threshold for data collection remains high. The relationship between time and vitality, as well as the underlying principles governing changes in urban vitality over time, remain unclear.

Thus, this study aims to elucidate the temporal dynamics of urban vitality. Previous research has highlighted significant disparities in the vitality of urban public spaces between day and night, as well as weekdays and weekends [11,13]. These spatiotemporal changes exhibit cyclical and periodic patterns and are closely linked to human activities [14,17]. Through observations of urban life, hypotheses regarding the relationship between urban temporal vitality and crowd activity were formulated. Building upon a review of past methodologies and a framework of hypotheses, the parameters and algorithms for dynamic vitality assessment were developed. Finally, the hypotheses were validated by comparing predicted values of public space vitality in case urban areas with measured values.

### 2.1. Hypothesis

The following hypothesis investigates the influence of crowd preferences and schedules on the utilization of urban spaces during specific time periods. It suggests that crowd behaviors and routines play a substantial role in shaping the usage patterns of various spaces.

- H01: The usage of a space during a particular period is influenced by crowd preferences.

Different functional spaces typically attract distinct user demographics. For instance, children are commonly found in children's park, while rarely seen in nightclubs.

Moreover, the primary users of the same space can vary across different times. For example, a café may primarily cater to office workers purchasing breakfast during early weekday mornings, whereas on weekend afternoons, it may be frequented by families gathering for leisure.

- H02: Specific crowds tend to adhere to predetermined schedules when utilizing urban spaces.

Demographic groups with similar age ranges and employment statuses typically maintain consistent routines. Although individual behavior may vary at the micro level, the population of a city block generally adheres to crowd patterns. For instance, full-time employees in Adelaide typically commence work between 8 and 10 am, take a lunch break between 12 noon and 2 pm, and conclude work between 4 and 6 pm. Consequently, it is unlikely for them to visit a restaurant at 3 pm on a weekday. Similarly, high school students, while fond of playing basketball, are unlikely to be present at the neighborhood basketball court during school hours. Hence, the likelihood of a specific crowd appearing in a particular location during a specific period can be inferred based on crowd preferences and schedules.

### 2.2. Specific Parameters

The next two parameters, Crowd and Frequency, stem from the aforementioned hypotheses. They outline the method of categorizing urban residents into crowd groups and predicting their activity frequencies in urban space.

- Crowds

Age and occupational characteristics significantly influence people's social interaction behavior patterns in urban spaces over time. For instance, during weekday working hours, full-time employees typically remain in their office positions, while individuals engaging in outdoor activities are often those not currently employed. Similarly, young children are rarely seen on the streets at night as they tend to sleep early.

Regarding crowd grouping methods, individuals in urban public spaces can be categorized based on age and employment status, and group rates can be calculated using local data statistics. Seven representative demographic profiles can be identified: young children, teenage students, unemployed individuals (including stay-at-home parents), full-time/part-time workers, people employed but not at work, and retired individuals (refer to Table 1). This classification can be further adjusted and refined as needed based on specific circumstances. For instance, if residents of a certain city commonly own dogs and the studied urban area includes a dog park, then an indicator for dog ownership should be included to refine the crowd classification.

**Table 1.** Common list of crowds' categories for urban public space.

| Crowd | Normal Age Range | Available Time in Public Spaces |
|---|---|---|
| Retirees | >local retirement age | Any time, but limited by energy |
| Employed but away from work | legal working age~retirement age | Available any time |
| Employed full-time | legal working age~retirement age | Available except standard working hours. |
| Employed part-time | legal working age~retirement age | Available outside working hours |
| Unemployed | legal working age~retirement age | Available any time |
| Teenager students | 5~legal working age | Available except during school hours |
| Young children | 0~4 | Near noon to afternoon |

- Frequency

The frequency of use is the most significant characteristic of functional spaces utilized by different groups of people, which can be deduced from the typical schedule of specific crowds. For instance, considering the average full-time employee in Adelaide, they typically work from 9 am to 6 pm on weekdays, with meal breaks and rest periods at noon,

commuting time to and from work, weekends off, and a willingness to engage in activities in public spaces. Building upon this assumption, a 24-h space appearance map was developed and delineated for both weekdays and weekends (refer to Table 2).

**Table 2.** Diagram of how often a typical space is used during the week of full-time workers in Adelaide. Blue means stay at home, green means stay in office and yellow means can appear in public urban space.

| Public Appearance | 00~08 | 08~10 | 10~12 | 12~14 | 14~16 | 16~18 | 18~20 | 20~22 | 22~24 |
|---|---|---|---|---|---|---|---|---|---|
| weekdays | House | Traffic | Work | Lunch | Work | Work | Traffic | House | House |
| weekends | House | Leisure | Leisure | Leisure | Leisure | Leisure | Leisure | Leisure | House |

For a specific public space, an activity frequency map for a particular type of crowd can be generated based on their schedule of appearance. If a specific crowd is expected to be active in the space at a given time, their probability of occurrence is designated as "1"; otherwise, it is marked as "0". Subsequently, the frequency of this particular crowd appearing in the space during that period can be quantitatively documented. By assigning weights to the frequencies of all crowds based on population proportions, the overall frequency of urban residents appearing in the space can be estimated.

## 3. Algorithm Construction

### 3.1. Indicators

The physical built environment, accessibility of the surroundings, and residents' characteristics influence the urban vitality of a place at a given time (Figure 2). In light of previous indicator systems and the crowd-frequency framework, a new indicator system for the dynamic time algorithm of urban public space vitality has been proposed.

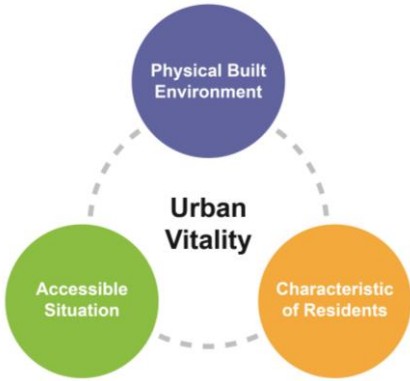

**Figure 2.** Factors that influence urban vitality.

### 3.1.1. Indicator System of Past Studies

Table 3 reviews five urban vitality studies' indicator systems. There is a lot of duplication and overlap between different indicator systems. In terms of physical built environment, these methods generally include information on space use, openness, landscape (greening) level, urban form, surrounding building functions and openness, etc. In terms of accessible situation, vehicle accessibility, bus accessibility, and walking accessibility are generally concerned. In terms of the characteristics of surrounding residents, they basically only focus on the population situation.

**Table 3.** Review of Indicator system of past research.

| Name and Date | Indicators | | Tools |
|---|---|---|---|
| Ye and Nes, 2014 [18] | Street-network configuration | | Space syntax |
| | Building density and types | | Space matrix |
| | Functional mixture | | Mixed use index (mxi) |
| | Other features | | |
| Li, et al, 2022 [15] | Street width | | GIS analysis of road data |
| | Greenery and openness and transparency | | Semantic segmentation of SVI |
| | Commercial density | | GIS analysis of POI |
| Li et al, 2022 [20] | Neighborhood attributes | Population density | Official statistics |
| | | Community age | Kriging method |
| | | Housing price | Kriging method |
| | Urban form | Floor-area ration | |
| | | Open space | |
| | | Intersection | |
| | | Road density | |
| | | Sidewalk percentage | |
| | | Streetlights | |
| | Facilities and land use | Food | POI data |
| | | Life service | POI data |
| | | Shopping | POI data |
| | | Lodging (HOT) | POI data |
| | | Transit stops (Bus) | POI data |
| | | Leisure | POI data |
| | | Tourist Attraction | POI data |
| | | Workplace | POI data |
| | | Land use mix | residential proportion |
| | Location | Distance to river | GIS |
| | | Distance to commercial | GIS |
| | | Distance to park | GIS |
| | | Distance to bus-stop | GIS |
| | | Distance to subway | GIS |
| | | Distance to leisure | GIS |
| | | Distance to plaza | GIS |
| | Landscape | NDVI | Landsat images |
| | Accessibility | Integration | SPACE SYNTAX |
| Guo et al, 2022 [19] | Indoor space next to the place | Spatial Social interaction coefficient (depends on function) | interaction ball |
| | | Area of indoor space | |
| | | Openness of buildings | |
| | Accessibility | Public Traffic | GIS |
| | | Walkability | GIS |
| | Outdoor public space | Spatial Social Interaction coefficient (depends on function) | interaction ball |
| | | Outdoor attraction points | |
| | Negative factors | Trashcan | |
| | | Others | |
| Lu et al, 2019 [21] | Social-economic data | Population | |
| | | House price | |
| | Compactness | Area | |
| | | Richardson compactness index | |
| | POI mixed use | Entropy | |
| | Accessibility | Density of bus stations | |
| | Density | Floor area ratio | |
| | | Building density index | |
| | | Road density index | |
| | Landscape | Green Coverage Index | |

### 3.1.2. New Indicator System

The new indicator system (Table 4) was designed by considering reviews and hypotheses about crowds and frequency. In selecting indicators, efforts were made to avoid redundancy and to prioritize simplicity and low acquisition cost. This involved screening out overlapping parameters and opting for indicators that exhibit similar characteristics but are simpler to acquire. Alongside the incorporation and utilization of common indicators from previous iterations, new factors such as crowds, frequency, and space for access were also integrated into the system.

**Table 4.** The indicator system of dynamic prediction algorithm of urban public space vitality.

| Factor Type | Indicators | |
| --- | --- | --- |
| Physical built environment | Urban public space | Spatial social interaction coefficient (depends on function) |
| | | Attractiveness of outdoor landscape and facility |
| | Building next to the public urban space | Area of ground floor indoor space open to public |
| | | Spatial social interaction coefficient (depends on function) |
| | | Openness of buildings |
| | Negative factors | Trashcan |
| | | Homeless |
| | | Others |
| Accessible situation | Vehicle accessibility | Reachable by car |
| | | Distance to parking lot |
| | Public transport system accessibility | |
| | Walking accessibility | |
| | Access factor, conditions for the urban space entry | |
| Characteristics of residents | Crowds | Age |
| | | Occupation and employment status |
| | Frequency schedule of residents | |

### 3.2. Algorithm and Formula

This algorithm model first calculates each parameter of the three types of factors separately.

Physical built environment factors:

$$F_B = lg \left( \sum_{a=1}^{a} (lg\,AREA_{in-a} \times INTERACT_{in-a} \times e^{OPENNESS}) + \sum_{b=1}^{b} (e^{ATTRACTION} \times INTERACT_{out-b}) - NEG) \right) \quad (1)$$

in the formula, $F_B$—the factor of physical built environment at a certain time.

$Area_{in}$—area of indoor space.

$INTERACT_{in}$—interaction coefficient of building space next to the place, depends on mean function and measured according to the interaction ball.

$OPENNESS$—openness coefficient for different functions in indoor buildings, values 0, 0.5, and 1 correspond to closed, semi-open, and fully open respectively.

$INTERACT_{out}$—outdoor interaction coefficient, depends on mean function and measured according to the interaction ball.

$ATTRACTION$—comprehensive attraction level of greening and public activity facilities in outdoor spaces takes an integer from 0 to 2.

$NEG$—internal and external negative factors.

Accessible situation factors:

$$F_A = Accessibility_{auto} \times Accessibility_{walk} \times Access\,factor \quad (2)$$

in the formula, $F_A$—the factor of accessible situation at a certain time.

*Accessibility<sub>auto</sub>*—accessibility factor of the vehicle of the place.

*Accessibility<sub>walk</sub>*—walking accessibility factor of the place.

*Accessfactor*—space access factor, whether there are conditions for space entry (such as membership or consumption); completely free is 4, quasi-free is 3, has potential consumption requirement is 2, fully charged / membership is 1.

Residents' characteristics factors:

$$F_C = lg\left(\sum_{m=1}^{\infty} GroupRate_m \times Frequency_m\right) \tag{3}$$

in the formula, $F_C$—the factor of residents' characteristics at a certain time.

*GroupRate*<sub>m</sub>—the proportion of crowd m in the regional population, calculated based on the local population age structure and employment situation.

*Frequency*<sub>m</sub>—the frequency of space usage of crowd m at a certain time, determined by the schedule of the typical image of crowd m. If crowd m can be active, the frequency value is "1", if not, the frequency value is "0".

Finally, the factors related to the physical built environment, accessibility, and residents' characteristics are multiplied to derive the prediction results of the urban vitality dynamic model, denoted as $\lambda$. Following the multiplication of all numerical products, the calculation result is logarithmically transformed to yield a more concise outcome.

$$\lambda = lg(F_C \times F_A \times F_B) \tag{4}$$

in the formula, $\lambda$—the predicted value of social spatial interaction of a space at a certain time.

The sorted algorithm formula of dynamic prediction of urban public space vitality is as follows:

$$\lambda = lg\left(\sum_{m=1}^{\infty} GroupRate_m \times Frequency_m\right) \times (Accessibility_{auto} \times Accessibility_{walk} \times Accessfactor)$$
$$\times lg\left(\sum_{a=1}^{a}(lg\, AREA_{in-a} \times INTERACT_{in-a} \times e^{OPENNESS}) + \sum_{b=1}^{b}(e^{ATTRACTION} \times INTERACT_{out-b}) - NEG)\right) \tag{5}$$

Among the parameters, GroupRate, Frequency, Accessibility<sub>auto</sub>, Accessfactor, OPENNESS, and NEG will vary over time. Different $\lambda$ values will be generated according to the calculation formula of the urban vitality dynamic prediction model at different times. The granularity of prediction in the time dimension is enhanced compared to previous research methods.

To handle variations in dimensions among model elements, streamline data processing, and mitigate the impact of outliers, the Z-Score normalization method is used [22,23]. This method ensures accuracy while retaining pertinent information and reducing sensitivity to outliers [24]. The processed data conforms to a standard normal distribution with a mean of 0 and a standard deviation of 1. To simplify comprehension and ensure positive values, data are collectively incremented by 'k'. The calculation formula for data standardization is as follows:

$$x_{new} = \frac{x - \mu}{\sigma} + k \tag{6}$$

in the formula, $x_{new}$—Standardized results.

$x$—original values in different dimensions.

$\mu$—the mean of the original data.

$\sigma$—the standard deviation (std) of the original data.

$k$—a positive integer, contingent upon data structure.

This normalization procedure will solely target parameters exhibiting significant mean and variance, such as area. Given that "0" and "1" in "frequency" signify whether an activity is occurring, standardizing the "frequency" element is unnecessary. Moreover, the values allotted to the factors of openness and influence are integers within the range of 4, rendering standardization unnecessary for these factors.

## 4. Prediction Framework Verification and Further Development through a Case Study

### 4.1. Verification Flow

The process of verification through a case study involves several steps (Figure 3). Initially, an urban zone is selected and divided into small blocks. The new dynamic model is then utilized to predict the results for each period of these blocks on weekdays and weekends. Subsequently, suitable dates are selected to gather street view data (short video and photo) for the selected blocks, and cross-sectional flow counting of spatial vitality [25] for each period on the sampled dates. The two sets of predicted and observed data are then processed into comparable indicators [24,26]. Finally, the reliability of the algorithm prediction model can be verified by evaluating and comparing these two sets of data.

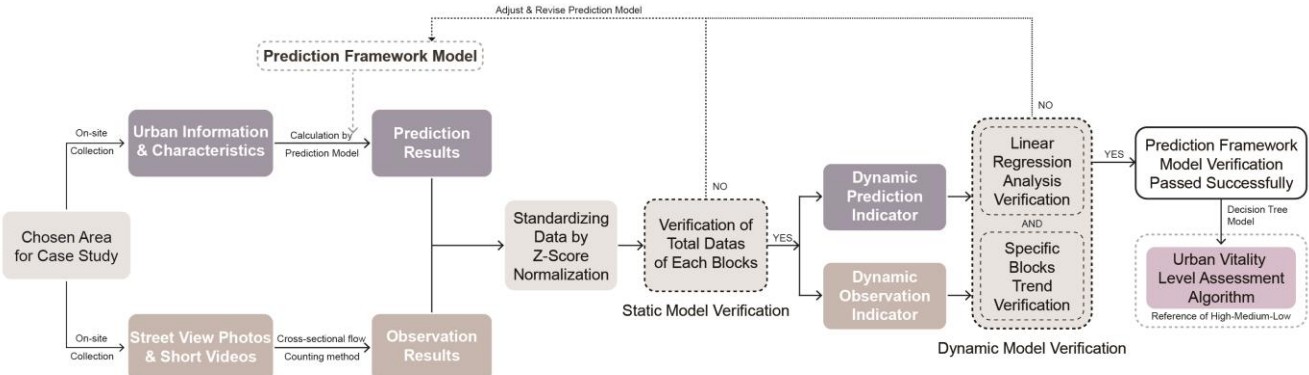

**Figure 3.** Working Flow Chart.

### 4.2. Selection of Case Area for Experimental Verification

#### 4.2.1. Adelaide Roundel Mall Block

For selecting urban zones for case studies, an area that is generally more active, with noticeable variations in activity levels across different time periods is suitable. Previous studies mostly chose bustling commercial districts or residential areas in city centers [19,25,27,28]. This ensures the collection of a sizable amount of data with significant variations, which is more conducive to validation.

Adelaide, the capital of South Australia, stands as the fourth largest city in Australia and has earned the title of "the most livable city in Australia" numerous times [29]. The central urban area of Adelaide encompasses Adelaide CBD, Green Ring, and North Adelaide. Rundle Mall Block, situated in the northeast of Adelaide CBD, serves as a quintessential example of an Australian urban block (Figure 4a). Rundle Mall finds itself adjacent to Adelaide University to the north, the headquarters business block to the south, a red-light district to the west, and Hindmarsh Square to the southeast, solidifying its status as one of South Australia's most beloved destinations (Figure 4b). Rundle Mall spans approximately 255,300 square meters and primarily functions as a hub for commercial retail, dining, and small-scale offices.

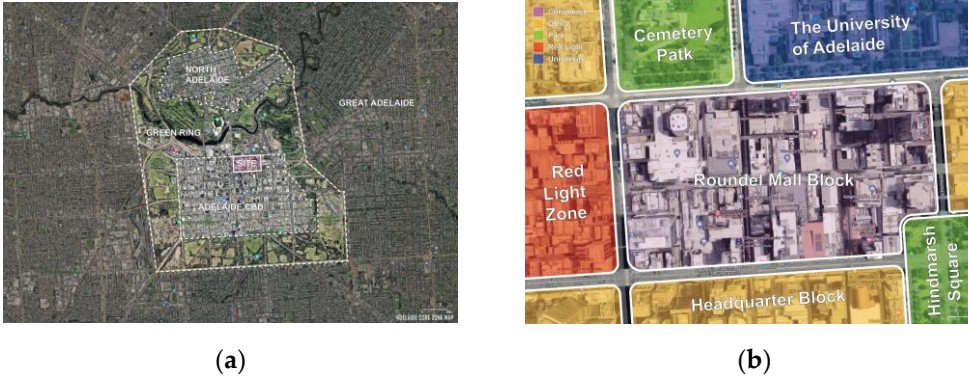

(**a**)         (**b**)

**Figure 4.** Location of Adelaide Rundle Mall Block: (**a**) Adelaide Core Zone Map; (**b**) Rundle Mall Block study area.

During the field trip, it was noted that while the overall urban vitality of the block is high, each area's vitality varies across time and space. Movement fluctuates between day and night and between different spaces, though not entirely synchronized (Table 5). Fewer individuals are present in the early morning and at night, while more are seen during noon and afternoon. Leisure hours see more foot traffic on inner pedestrian pathways, whereas traffic times witness more activity on outer walking paths. Weekends and holidays maintain high spatial vitality, with variations depending on location. From lunchtime until evening, the pedestrian street is bustling, while surroundings remain relatively quiet. Overall, the block's conditions are conducive to verifying dynamic algorithm models.

**Table 5.** Uneven spatial and temporal distribution of spatial vitality in Adelaide Roundel Mall.

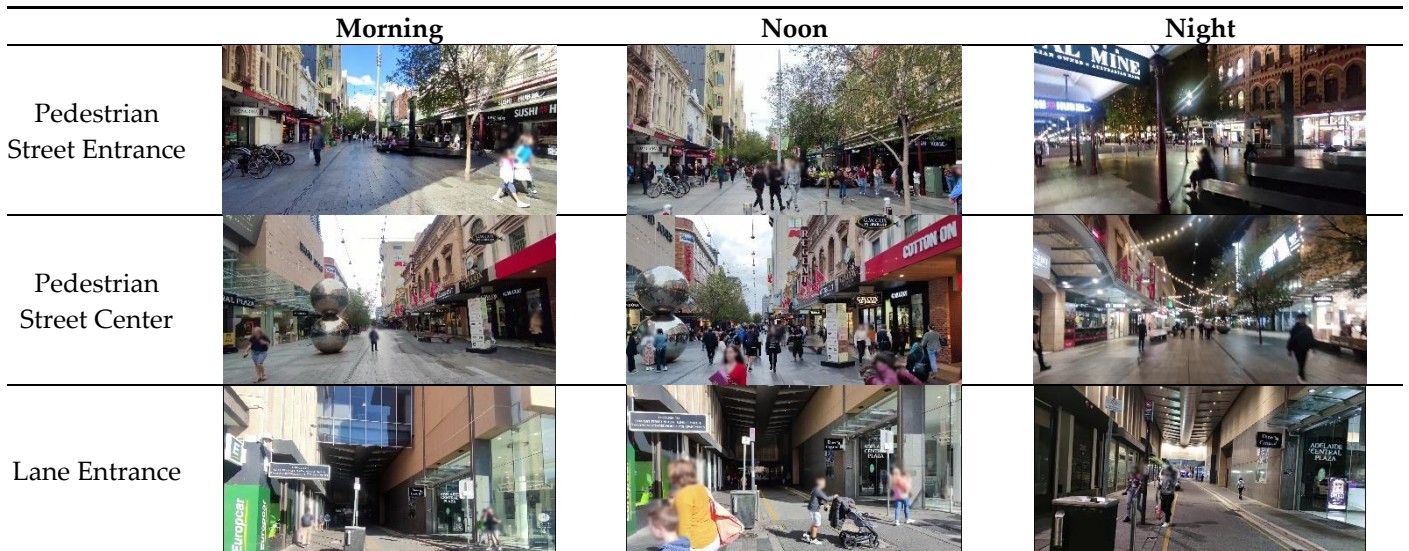

| | Morning | Noon | Night |
|---|---|---|---|
| Pedestrian Street Entrance | | | |
| Pedestrian Street Center | | | |
| Lane Entrance | | | |

### 4.2.2. Suite Division and Numbering

The public urban space within the Rundle Mall Block was subdivided into 98 small blocks (Figure 5), delineated by both physical environmental boundaries and the functional characteristics of adjacent buildings. To enhance verification efficiency and reduce the costs associated with field data collection, this study meticulously assessed spatial attributes and distribution locations. Consequently, 10 blocks, as depicted in Figure 6 and Table 6, were selected for on-site data collection.

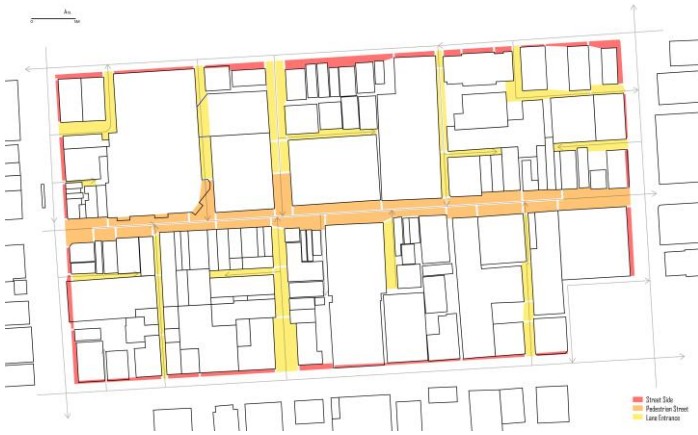

**Figure 5.** Site division of Adelaide Roundel Mall.

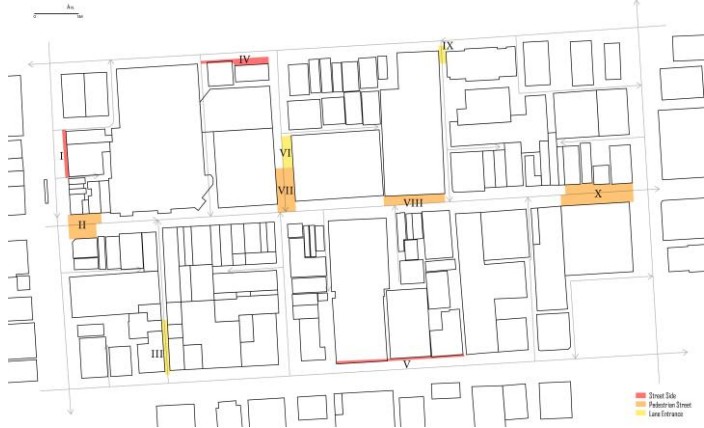

**Figure 6.** Selection of study area.

**Table 6.** Selected block location.

| Block Num | Location | Nearby Building Function |
|---|---|---|
| I | Street Side | Bank |
| II | Pedestrian Street Entrance | Retail |
| III | Lane Entrance | Retail |
| IV | Street Side | Apartment and Club |
| V | Street Side | Shopping mall |
| VI | Rest in Lane | Retail |
| VII | Pedestrian Street Node | Retail |
| VIII | Pedestrian Street | Shopping mall |
| IX | Lane Entrance | Shopping mall |
| X | Pedestrian Street Entrance | Retail |

*4.3. Obtaining and Calculating Each Parameter in the Prediction Model*

4.3.1. Crowds and Frequency Analysis

According to the framework of the new dynamic model, the initial step involves analyzing and determining the proportion of representative crowds expected to frequent the area. As this area constitutes the core of the entire city, its users are not limited to residents of Adelaide CBD but encompass individuals from the broader Great Adelaide region. Population data utilized in this study were sourced from the 2021 Census All persons Quick Stats for Great Adelaide, released by the Australian Bureau of Statistics (Figure 7) [30]. Through an examination of factors such as age and employment status (refer to Figure 6), the final statistical outcome regarding the ratio of crowds is derived (refer to Table

7). By combining population data with the fundamental functional planning of the area, the following conclusions can be drawn: (1) The age demographic of primary area users is highly diverse. (2) On weekdays, the area is expected to primarily attract office workers and nearby students, while weekends are anticipated to draw families, individuals seeking leisure activities, gatherings with friends, and shoppers.

**Table 7.** Crowds' data and Block II Frequency Estimation Results.

| | Frequency of Block II | | | | | | |
|---|---|---|---|---|---|---|---|
| **Days** | | Weekdays | | | | | |
| **Crowds** | **Proportion** | **8~10** | **10~12** | **12~14** | **14~16** | **16~18** | **18~20** |
| Retirees | 16% | 0 | 0 | 1 | 1 | 1 | 0 |
| Employed but away from work | 4% | 1 | 1 | 1 | 1 | 1 | 0 |
| Employed full-time | 40% | 0 | 0 | 1 | 0 | 1 | 0 |
| Employed part-time | 27% | 1 | 0 | 1 | 1 | 1 | 0 |
| Unemployed | 7% | 0 | 0 | 1 | 1 | 1 | 1 |
| Teenager students | 4% | 0 | 0 | 1 | 1 | 1 | 0 |
| Young children | 2% | 0 | 0 | 1 | 1 | 1 | 0 |
| Overall | 100% | 0.036 | 0.036 | 0.998 | 0.328 | 0.998 | 0.073 |
| **Days** | | weekends | | | | | |
| **Crowds** | **Proportion** | **8~10** | **10~12** | **12~14** | **14~16** | **16~18** | **18~20** |
| Retirees | 16% | 0 | 0 | 1 | 1 | 0 | 0 |
| Employed but away from work | 4% | 1 | 1 | 1 | 1 | 1 | 0 |
| Employed full-time | 40% | 0 | 0 | 1 | 1 | 0 | 0 |
| Employed part-time | 27% | 0 | 0 | 1 | 1 | 0 | 0 |
| Unemployed | 7% | 0 | 0 | 1 | 1 | 1 | 1 |
| Teenager students | 4% | 0 | 1 | 1 | 1 | 1 | 0 |
| Young children | 2% | 0 | 0 | 1 | 1 | 0 | 0 |
| Overall | 100% | 0.036 | 0.074 | 0.998 | 0.998 | 0.147 | 0.007 |

(**a**)

(**b**)

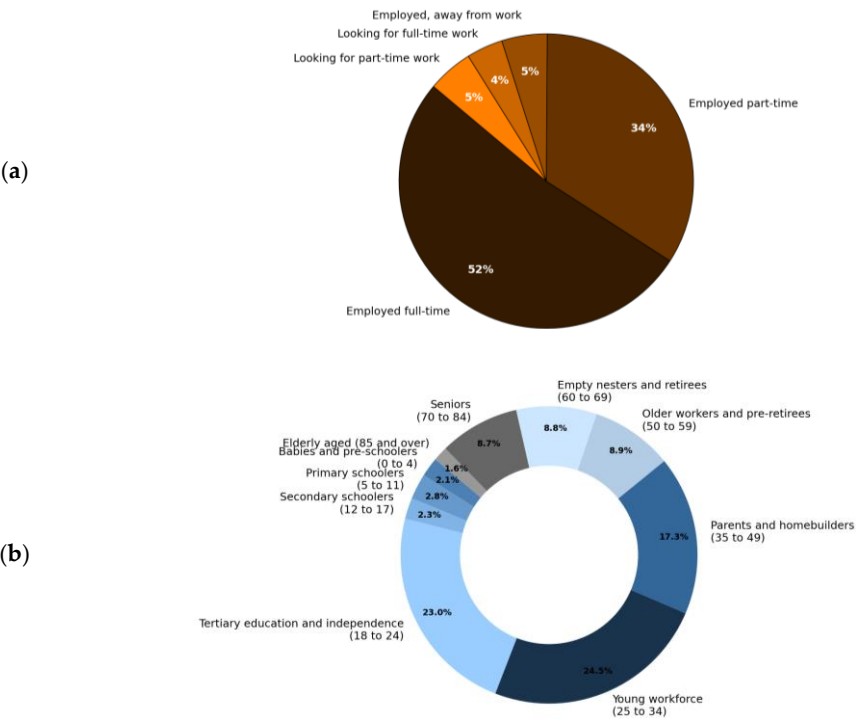

**Figure 7.** Population data of Adelaide: Persons in the Adelaide labor force (**a**), Adelaide population age group (**b**) [30].

Using functional attributes, crowd behavioral characteristics, and the proportion of each urban space, the frequency of each block during different time periods is estimated. The table below provides an example using Block II to illustrate specific frequency estimation results based on crowd proportions and "0" and "1" probability methods.

4.3.2. Collection of Other Parameters in the Prediction Model

Values are obtained in the following ways:

- Area—functional area. The results are obtained by counting the building area and site area in each spatial unit.
- Space access coefficient—assigned based on the openness of each functional space inside and outside the space unit. Completely free is 4, quasi-free opening requirement is 3, potential consumption requirement is 2, fully charged is 1.
- Interaction coefficient—mainly determined based on the land use properties of the space unit, combined with the functional characteristics of indoor and outdoor spaces to assist judgment, and finally determined based on the interactive sphere model.
- Attraction coefficient—determined by the type and level of landscape/activities in the space. According to the location and scale of the attraction point, assign values from 0 to 3, respectively. The higher the value, the greater the influence.
- Auto accessibility—comprehensive calculation based on the distance to public transportation stops. There are nine bus stations, and three train stations around the research plot.
- Walking accessibility—comprehensive calculation of walkable area.
- NEG—spatial external negative factors. First, conduct on-site research to determine the number of negative impact points in the site and make statistics. A large trash can is worth 1, a small trash can is worth 0.5, and a homeless person is worth 1. Then the negative records of each spatial unit are accumulated and obtained.

*4.4. Field Observation Data Collection and Processing*

The researchers began their investigation by observing the behavior of individuals and crowds in Rundle Mall over the course of one month, aiming to understand the patterns of crowd behavior. In previous studies concerning the collection of human behavior data in public spaces, researchers conducted on-site street view image collection for each study area, ranging from 1 to 6 days, distinguishing between weekdays and weekends [25,28,31,32]. These studies meticulously controlled factors such as weather and temperature, deliberately selecting days with mild weather conditions. Hence, to mitigate the influence of severe weather and natural calamities like rain, hail, heatwaves, wildfires, high solar radiation, low temperatures, and strong winds on urban activities, the researchers deliberately chose Adelaide's autumn, characterized by more pleasant and stable weather conditions (with an average temperature of 12~22 °C), for on-side data collection. Specifically, they collected street view data spanning from 27 April to 30 (2 weekdays and 2 weekends) in 2023. Additionally, holidays and public vacations were intentionally avoided to ensure that the observed behavior was representative of typical urban dynamics on normal days. Data were collected every two hours during 8:00~20:00, using both short videos and photographs to track the presence of individuals within the designated area. Then, according to the cross-sectional flow counting method [25], urban vitality was calculated, and the averages were taken based on weekdays and weekends. Subsequently, these calculated averages were transformed using natural logarithms and standardized [24,26]. This processed data served as an indicator of urban vitality for the specified time and location.

*4.5. Data Analysis and Comparison Verification*

Given the potential presence of outliers in the statistical data collected during this study, a Z-Score normalization method was employed for preprocessing. This approach aims to retain characteristic information within the outliers while reducing sensitivity to

their impact. The calculation formula for data normalization is consistent with the previously mentioned Formula 6 and will not be reiterated here.

### 4.5.1. Static Model Results Validation

Firstly, the dynamic parameters introduced by the crowd-frequency hypothesis, associated with the temporal dimension, are disabled. The predicted results are then compared with the average on-site observations throughout the day. Refer to Figure 8, the relative magnitudes of predicted values at each site closely correspond to the trends observed in the overall daily cumulative data. However, it is notable that the predicted values for the VII block show a significant deviation from the actual values, overestimating them. This block serves as a pedestrian street node where food gatherings are frequently held, but its opening hours are much shorter than those of regular commercial areas. Consequently, the static prediction of the overall vitality level for this block tends to be much higher than the actual level throughout the day. This further highlights the necessity of developing a dynamic prediction model. Despite this discrepancy, the validation process demonstrates robust consistency, confirming the successful establishment of the static part of methods and indicators of the model.

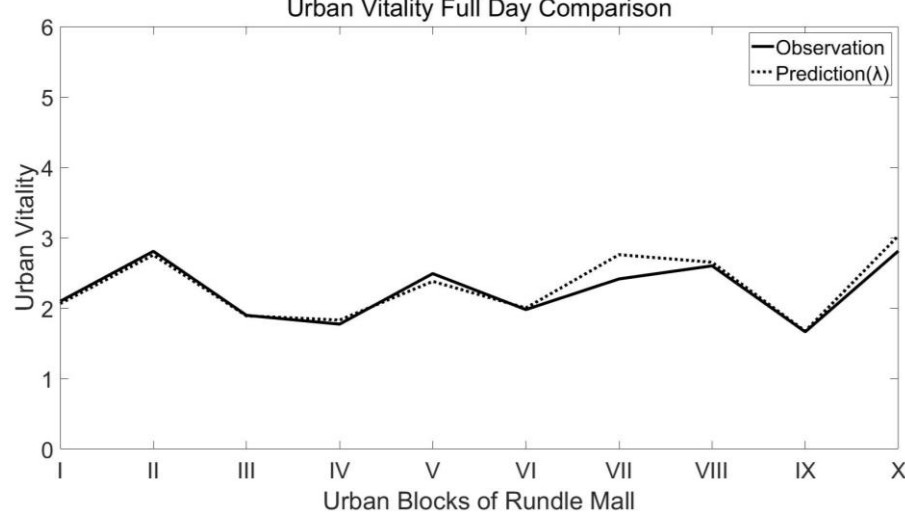

**Figure 8.** Comparison of static model and on-site observation.

### 4.5.2. Dynamic Model Results Validation

Building upon the certified static model, further validation of the crowd-frequency hypothesis is conducted. To ensure comparability of two sets of values with different units, both sets of values have undergone standardization [26] and are presented as unitless indicators for comparison. Firstly, through linear regression (Table 8), we analyzed the correlation between the predicted results of 10 sites during six time periods on weekdays and weekends, and the on-site observation results. The calculated result of $R^2$ = 0.847572 > 0.8 [33–35], with a *p*-value = 0.080326 < 0.1 [36,37], indicates significant correlation, demonstrating successful overall model fitting.

**Table 8.** Linear regression analysis.

| Regression Statistics Parameters | Multiple R | $R^2$ | *p*-Value |
|---|---|---|---|
| Result | 0.920637 | 0.847572 | 0.080326 |
| Trusted range | ---- | >0.8 [33–35] | <0.1 [36,37] |

Next, the data variation for specific blocks is verified. This step aims to confirm that the data aligns with common science and prevents the linear regression calculation from

being based on erroneous results. The following Table 9 presents the comparison between the dynamic model's predictions and on-site observations for three types of blocks. Each type of block shows an example, with weekdays on the left and weekends on the right. In Table 9, solid lines represent the observed value indicators obtained from on-site data collection, while dashed lines represent the predicted value (λ) indicators calculated based on the dynamic algorithm. The results illustrate the fluctuations in urban vitality levels for each of the three block types during weekdays and weekends from 8:00 to 20:00, in two-hour intervals. The trends align well, accurately corresponding to peaks and troughs. In the overall urban vitality ranking, the Pedestrian Street displays first, followed by the Street Side, while the Lane exhibits the last. Additionally, there is a noticeable increase in vitality starting from noon in each day and block. These findings are aligned with common knowledge [n]. The validation of the dynamic model for specific blocks has also been successful.

**Table 9.** Examples of comparison of urban vitality dynamic predictions and on-site observations of 3 types of blocks.

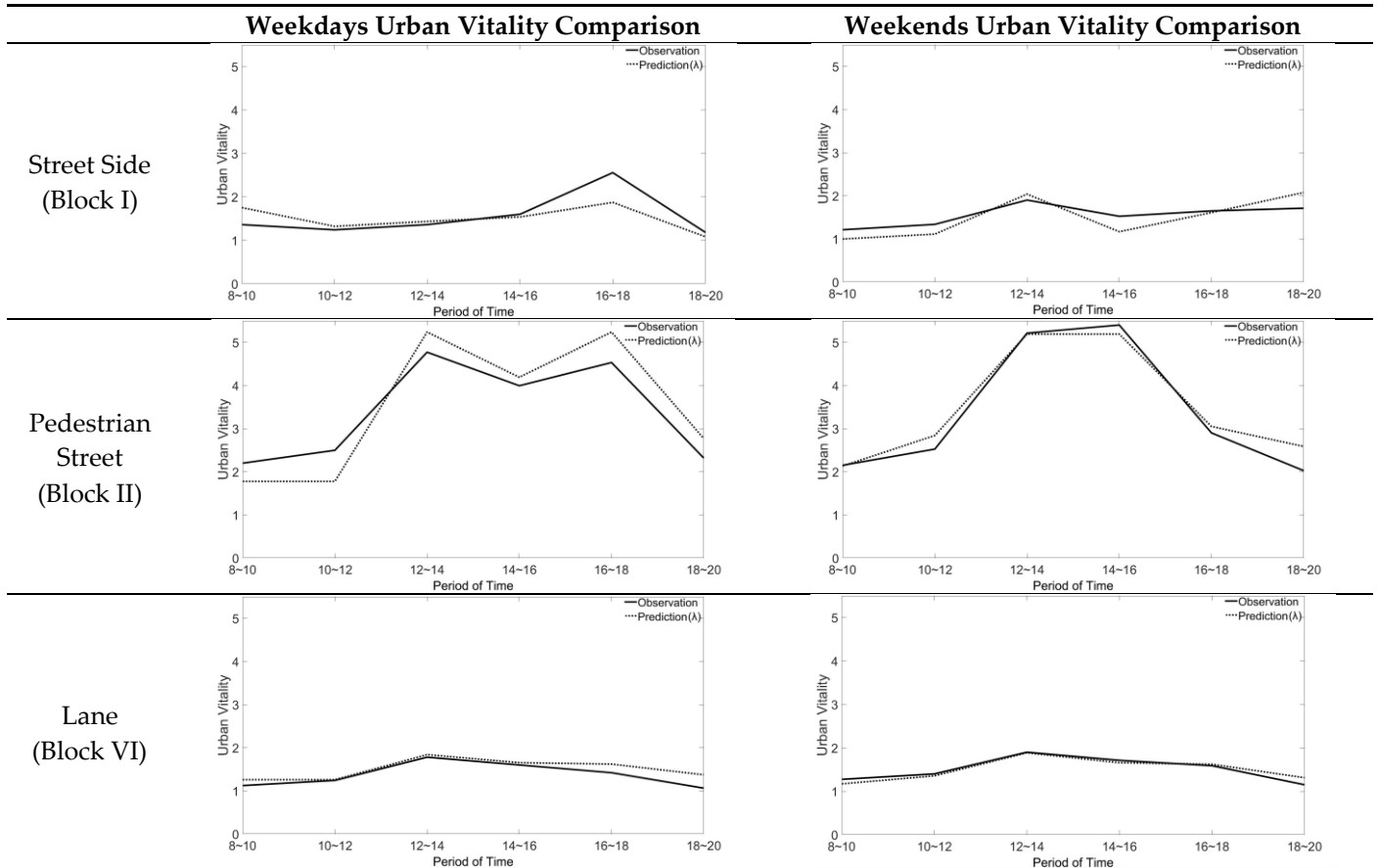

### 4.6. A Preliminary Vitality Level Prediction Program Based on Decision Tree Model

Section 4.5 has demonstrated the effectiveness of the dynamic urban vitality prediction framework. However, since the primary goal of predicting urban vitality is to assess the level rather than the exact values, the numerical indicators output by the current model might confuse users regarding the interpretation of the high-low levels of public space vitality. Although Table 9 illustrates that the prediction model accurately captures the trend of vitality changes in urban public spaces, it also indicates that there are differences in the specific predicted values, with variations in magnitude.

To tackle this issue, this study utilized a decision tree model [38,39] for machine learning on the aforementioned data outcomes. An algorithmic program was created to predict urban vitality levels based on inputted basic information and temporal conditions

of specific spaces. The program combines parameters of dynamic algorithmic formulas into three sets of independent variables ($F_A \times F_B$, $F_C$, and period×weekdays/weekends) and establishes natural breakpoints [40] for the target values (c = 3). Employing a simulation with test_size = 0.34, the algorithm achieves an F1_micro score of 0.93878, surpassing the reliability threshold [41]. The output of this algorithm assigns integer values of 1, 2, and 3, corresponding to low, moderate, and high levels of urban vitality, respectively. This approach aims to improve user understanding and utilization of the predictive model.

However, it should be noted that the concept of vitality levels is inherently relative. The method used to label vitality levels is based on on-site observed indicator data, categorized using a natural breakpoint approach according to numerical values (c = 3). The current training dataset is derived from four ordinary dates in Adelaide CBD, providing a preliminary reference for urban vitality level predictions based on them. However, it is acknowledged that different regions and cultural backgrounds may interpret vitality levels differently. For instance, what might be considered "very busy" in Adelaide might appear only "moderately busy" in a densely populated cities like Hong Kong. Nevertheless, there is generally a consensus on the definition of low activity. Considering the fundamental aim of developing this prediction framework and algorithm is to mitigate the occurrence of "ghost streets" during the design phase, it remains valuable. In future, researchers will gather more data from cities with diverse geographical and cultural backgrounds to enhance the predictive model's reference value.

## 5. Discussion

Firstly, the Dynamic Prediction Framework for Urban Public Space Vitality has been successfully validated in Adelaide CBD (Table 8), confirming the hypothesis regarding the time-dependent influences of crowd-frequency in urban public spaces. During the actual computations, it was found that among the six dynamic parameters in the algorithm formula proposed in Section 3.2, the two parameters of crowd-frequency, GroupRate, and Frequency, exhibit the greatest variability, followed by OPENNESS, which reflects the operational status of adjacent buildings. Accessibility$_{auto}$ and Accessfactor show minimal changes between 8 a.m. and 8 p.m., while NEG experiences a slight increase in the evening. Although the operational status of buildings is an objective condition for urban public spaces, it is ultimately influenced by the behavior of people within the buildings. Therefore, in terms of the time dimension, the changes in urban public space vitality are fundamentally based on the variations in resident behavior. Despite the complex nature of individual behaviors, the frequency of space usage by different crowds and the overall space usage frequency are predictable. Moreover, because frequency predictions can be adjusted based on local cultural customs and lifestyle patterns in different countries and regions, this forecasting method possesses universality.

Secondly, the outcomes of the new predictive model based on crowd frequency and temporal dimensions exhibit a high degree of consistency with the overall trends of real-world observations. Nevertheless, disparities persist in the level of alignment across different locations, likely due to biased utilization of population data. The validation process of the model used population data from the greater Adelaide area (2021) to estimate pedestrian flow in the block of Rundle Mall (2023), essentially providing an average value for the broader region while approximating specific values for smaller areas. Further refinement and revision of population data may be necessary to address these disparities.

However, the Dynamic Prediction Framework Model developed in this study also has limitations. Firstly, the analysis, discussion, and validation in this research only focused on regular weekdays and weekends, overlooking significant holidays. Events such as large-scale tourist groups, parades, and strikes were not taken into account. Additionally, the algorithmic model is only applicable under ideal weather conditions (clear skies, mild winds, and pleasant temperatures), while the impacts of extreme weather conditions such as thunderstorms, hurricanes, and heatwaves on people in urban public spaces were not considered. Furthermore, Adelaide, used for analysis, discussion, and validation, is

not a tourist city, and short-term travelers other than holiday travelers are rare on non-festival holidays. For tourist destination cities, the active population body is no longer the local permanent population but tourists. This leads to significant differences in crowd grouping, requiring more refined dynamic data on population. The applicability of this dynamic prediction model to predict urban vitality in public spaces in such tourist cities awaits further validation.

Furthermore, a preliminary predictive algorithm generated by decision-tree model offers designers predictions of urban vitality reference levels for design schemes. The fundamental parameters required for inputting into the urban public space vitality dynamic prediction framework are crowd frequency ($F_C$), physical environment ($F_B$), arrival situations ($F_A$), and time conditions (period, weekdays/weekends). These input data are straightforward and readily obtainable, offering a cost-effective and efficient means to optimize design vitality and urban planning solutions, thereby reducing waste and achieving sustainable high-vitality development. Although the current output results serve as reference values and the definition of vitality levels is based on four days on-site observations in Adelaide, in the future, as more data is collected from different areas and dates, the algorithm can become more refined.

## 6. Conclusions

Firstly, this study demonstrates that the theory of the dynamic model is supported by practical evidence. The vitality of urban public spaces is influenced by a combination of physical environment, traffic conditions, and residents' behavioral characteristics. The hypotheses proposed regarding crowd-frequency have been validated. Among these factors, the opening status of buildings adjacent to public spaces and the changing activity frequency of different groups of people over time have the greatest impact on dynamic vitality in the temporal dimension, exhibiting predictable patterns.

Secondly, the new predictive method and framework model, is entirely independent of whether the projects have been completed or not. Additionally, the dynamic model shows higher accuracy in its results than traditional static models. It can help prevent phenomena such as "ghost streets" during specific periods and promotes sustainable development in urban areas.

Additionally, a preliminary prediction program, proposed based on a decision tree model, can utilize foundational parameters set by the predictive model to predict high-medium-low reference levels of urban vitality. When applied in other countries and regions, critical interpretations of high-medium-low vitality levels should be made based on the differences in local culture and lifestyle compared to Australia.

In future research, authors will further explore classifying crowds in different types of urban areas to investigate the applicability of the dynamic prediction framework model. This may involve examining various urban environments, such as suburban housing neighborhoods and tourist towns, as well as other areas with unique functions or characteristics. Additionally, more social behavioral studies may be conducted to gain deeper insights into how people interact with urban environments and contribute to research on urban vitality.

**Author Contributions:** Conceptualization, Y.L. and X.G.; methodology, Y.L. and X.G.; Data collection & algorithm model built, Y.L.; writing—original draft preparation, Y.L.; writing—review and editing, X.G. All authors have read and agreed to the published version of the manuscript.

**Funding:** This research was funded by: Later Funded Projects of National Philosophy and Social Science Foundation of China, grant number No. 19FXWB026; Higher Education Research and Reform Project of Guangdong Province (No. HITSZERP19001); General Project of Stabilization Support Program of Shenzhen Universities (No. GXWD20201230155427003-20200822174038001); Key Laboratory of Cognitive and Personality, Ministry of Education, Chongqing, 400715; Shenzhen Education Science "14th Five-Year Plan" 2022 Annual Topic (No.: cgpy22018).

**Institutional Review Board Statement:** Not applicable.

**Informed Consent Statement:** Not applicable.

**Data Availability Statement:** The data presented in this study are available on request from the corresponding author. The data are not publicly available due to privacy.

**Conflicts of Interest:** The authors declare no conflict of interest.

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
