# Peer review of "A Dynamic Prediction Framework for Urban Public Space Vitality: From Hypothesis to Algorithm and Verification"

_sustainability, doi:10.3390/su16072846_

Round 1

Reviewer 1 Report (Previous Reviewer 1)

Comments and Suggestions for Authors

1.     Authors should specify whether they want to prevent issues such as "ghost neighborhoods”, “ghost blocks" or phenomena such as "ghost streets". Ghost neighborhoods were mentioned in the abstract, but actually the authors deal with ghost streets? So, please, be more specific.

2.     Authors should explain the vitality level of prediction of a public space during specific period.

3.     Please, improve Figure 6.

4.     Figure 7.: Author should label the x-axis and explain what it represents. 

5.     In the manuscript, there is Table 8. Please explain (in manuscript) the data shown in the table. What do the results (data) refer to? There is no discussion of the results.

6.  Table 9.:   Graphs in this table are not visible and clear, please improve Table 9.

Author Response

Response to Reviewer 1 Comments

1. Summary

Thank you very much for taking the time to review this manuscript. We have carefully considered your comments and have made significant revisions, particularly in sections 4 5 and 6 to of the manuscript. The modified sections have been highlighted in yellow. We appreciate your valuable feedback and believe that these revisions have strengthened the overall quality of the manuscript. Please find the detailed responses below and the corresponding revisions in the re- submitted files.

2. Questions for General Evaluation

Reviewer’s Evaluation

Response and Revisions

Is the content succinctly described and contextualized with respect to previous and present theoretical background and empirical research (if applicable) on the topic?

Can be improved

Yes, the content has been succinctly described and contextualized with respect to previous and present theoretical background and empirical research on the topic. The background section has been improved.

Are all the cited references relevant to the research?

Yes

Are the research design, questions, hypotheses and methods clearly stated?

Can be improved

There is expanded elaboration on these aspects

Are the arguments and discussion of findings coherent, balanced and compelling?

Can be improved

A flow chart has been added to section 4.1, and explanations have been added to section 4.4.

For empirical research, are the results clearly presented?

Can be improved

Improvements have been made in sections 4.5 and 4.6, with added explanations of the results.

Is the article adequately referenced?

Yes

Are the conclusions thoroughly supported by the results presented in the article or referenced in secondary literature?

Can be improved

The conclusion has been completely rewritten

3. Point-by-point response to Comments and Suggestions for Authors

Comments 1: [Authors should specify whether they want to prevent issues such as "ghost neighborhoods”, “ghost blocks" or phenomena such as "ghost streets". Ghost neighborhoods were mentioned in the abstract, but actually the authors deal with ghost streets? So, please, be more specific.]

Response 1: Thank you for pointing this out. I agree with this comment. Therefore, I have made the necessary revisions in the manuscript to address this inconsistency. In our study, we focus on the phenomenon of "ghost streets" rather than "ghost neighborhoods" or "ghost blocks". Given that our research primarily concerns the microscale, using consistent terminology is essential to avoid confusion. As per your suggestion, I have updated the text on line 10 and line 26 into "ghost streets", aligning with the concept mentioned in the 4.6 at line 457.

“[line 10: … aiming to prevent issues such as "ghost streets" and minimize resource wastage.

line 26: … the emergence of " ghost streets" in various new towns…]”

Comments 2: [Authors should explain the vitality level of prediction of a public space during specific period.]

Response 2: Thank you for pointing this out. We define the vitality level as the pedestrian flow, representing the density of people in the space. It's important to note that the concept of high or low vitality levels is inherently relative and is derived from natural breakpoints in the measured indicator data, which is a common approach in data segmentation. The reference values provided are based on on-site observations on typical dates in Adelaide CBD. However, it's well known that different cities, with their unique geographical and cultural backgrounds, may interpret vitality levels differently. For instance, what might be considered "very busy" in Adelaide could be seen as only "moderately busy" in densely populated areas like Hong Kong. Nevertheless, there's generally a consensus on what constitutes low activity. Considering the fundamental purpose of developing this predictive framework and algorithm, which is to reduce the occurrence of "ghost streets" during the design phase, it remains valuable. We agree that our previous expression and description of the algorithm values were not precise enough. It does indeed provide a reference for the urban vitality level in the capital city of South Australia. Future researchers will gather data from cities with different geographical and cultural backgrounds to enhance the reference value of the predictive model.

Comments 3: [Please, improve Figure 6.]”

Response 3: Agree. We have enlarged.

Comments 4: [Figure 7.: Author should label the x-axis and explain what it represents.]”

Response 4: Agree. We have modified.

Comments 5: [In the manuscript, there is Table 8. Please explain (in manuscript) the data shown in the table. What do the results (data) refer to? There is no discussion of the results.]”

Response 5: Agree. We have simplified Table 8, displayed only the key numerical values, and added an interpretation of the data.

Comments 6: [Table 9.:   Graphs in this table are not visible and clear, please improve Table 9.]”

Response 6: Agree. I We have redrawn the labels.

4. Response to Comments on the Quality of English Language

Point 1: English language fine. No issues detected

Response 1:    Thank you

5. Additional clarifications

none

Reviewer 2 Report (Previous Reviewer 2)

Comments and Suggestions for Authors

Thank you for taking in consideration my suggestions. In my opinion only four days for collecting data is not enough to predict results for all year. However, if you will continue the research for a longer period of time – at least 4 days a month, I will agree with the publication of this paper in this form.

Comments on the Quality of English Language

From the point of view of the English language, the article is written correctly, there are some smaller problems with some grammar correction, pictures, the necessary corrections were indicated in the attached document.

Author Response

Response to Reviewer 2 Comments

1. Summary

Thank you very much for taking the time to review this manuscript. We have carefully considered your comments and have made significant revisions, particularly in sections 4 5 and 6 to of the manuscript. The modified sections have been highlighted in yellow. We appreciate your valuable feedback and believe that these revisions have strengthened the overall quality of the manuscript. Please find the detailed responses below and the corresponding revisions in the re- submitted files.

2. Questions for General Evaluation

Reviewer’s Evaluation

Response and Revisions

Is the content succinctly described and contextualized with respect to previous and present theoretical background and empirical research (if applicable) on the topic?

Can be improved

Yes, the content has been succinctly described and contextualized with respect to previous and present theoretical background and empirical research on the topic. The background section has been improved.

Are all the cited references relevant to the research?

Can be improved

Adjustments have been made to improve their relevance. Additionally, new references have been added to further enhance the relevance and comprehensiveness of the literature review.

Are the research design, questions, hypotheses and methods clearly stated?

Can be improved

There is expanded elaboration on these aspects

Are the arguments and discussion of findings coherent, balanced and compelling?

Can be improved

A flow chart has been added to section 4.1, and explanations have been added to section 4.4.

For empirical research, are the results clearly presented?

Must be improved

Improvements have been made in sections 4.5 and 4.6, with added explanations of the results.

Is the article adequately referenced?

Can be improved

The number of references has been increased from 22 to 41, providing stronger support for the

Are the conclusions thoroughly supported by the results presented in the article or referenced in secondary literature?

Can be improved

The conclusion has been completely rewritten

3. Point-by-point response to Comments and Suggestions for Authors

Comments 1: [Thank you for taking in consideration my suggestions. In my opinion only four days for collecting data is not enough to predict results for all year. However, if you will continue the research for a longer period of time – at least 4 days a month, I will agree with the publication of this paper in this form.]

Response 1: Thank you for raising this concern. We have addressed this issue in the revised manuscript to provide clarity on our approach.

1. Regarding the limited duration of our on-site observation data collection, we acknowledge that urban vitality exhibits strong cyclical patterns (lines 42-46) . Therefore, we chose to validate our framework using data collected over one week as it suffices to capture these cyclical variations.

2. It's important to note that our predictive framework model is independent of the on-site observation data. The framework was constructed objectively, without reliance on the observed data. The selection of April for data collection was based on the pleasant and stable weather conditions in Adelaide during autumn, which better reflects the fundamental behavioral patterns of individuals unaffected by weather conditions. Other seasons could introduce external factors such as weather disturbances, which could impact the statistical results.

3. The decision to collect data for only four days (2 weekdays and 2 weekends) was determined based on the typical data collection practices in similar studies in recent years. Previous studies, such as Xu & Chen (2023) sampled every two hours on 2 weekdays in October for each study area; Chen et al. (2024) sampled every four hours on a weekday, a Saturday, and a Sunday for each case park scene; Mu et al. (2021) collected data for three weekdays and three weekends in April; Hu et al. (2024) sampled every hour on a weekday in July. They also utilized similar durations for data collection, which we deemed sufficient for initial validation.

4. Data collection can be challenging, and despite our efforts to collect data in May, November, and March of the previous and current years, we faced unexpected weather events such as heavy rain, hail, heatwaves, and temporary public events (food festivals, gaming gatherings), which disrupted our data collection efforts. Therefore, we only have full-day data available for these four days.

5. As a result, we believe that the data collected for initial validation is adequate. These four days of data were used to train a vitality reference level algorithm (4.6). Only this part is based on existing field data collection. It's just a byproduct of this predictive framework. We plan to collect data from more areas and dates in the future to further validate and refine our urban vitality level algorithm predictive model.

Comments 2: [Figure 6 Unclear]

Response 2: Agree. We have enlarged them.

Comments 3: [table 9 Unclear]

Response 3: Agree. We have redrawn the labels.

Comments 4: [line 343 heat waves]

Response 4: Agree. We change the mistake.

4. Response to Comments on the Quality of English Language

Point 1: Minor editing of English language required

Response 1:    We have carefully reviewed the manuscript and made minor adjustments to enhance the clarity and fluency of the English language. These revisions include refining sentence structures, improving word choices, and ensuring coherence throughout the text. We believe that these enhancements will contribute to a more polished and professional manuscript.

5. Additional clarifications

none

Reviewer 3 Report (Previous Reviewer 3)

Comments and Suggestions for Authors

The paper makes a significant contribution to urban public space research, emphasizing the importance of vitality in its planning. The context is set out in detail in the introduction. Identifying current issues such as underutilization in certain areas and the need for costly redevelopment adds significant relevance and importance to the research. The authors provides a clear and relevant definition of urban vitality, emphasizing the "intensity of people's concentration" and highlighting its importance in public space design. The authors provide a relevant definition of urban vitality, highlighting the "intensity of people's concentration" and stressing its importance in the design of public space. A review of the existing methods for evaluating urban vitality is presented, and their limitations are analyzed. Research methods - description of algorithms, parameters and calculation methods - are described in detail. The proposed method is verified in the presented case study. The limitations of the proposed algorithm are also highlighted.

Notes: Bibliographic references could be improved - even if the references presented support the paper, I feel that some expansion is necessary. Authors must pay more attention to the way of writing: why are some texts colored yellow? Some images are low resolution and do not compliment the research.

Considering the fact that the research is presented in detail, the conclusions do not fully support it. The Conclusions section seems more like a succinct enumeration of the main parts of the paper.

Author Response

Response to Reviewer 3 Comments

1. Summary

Thank you very much for taking the time to review this manuscript. We have carefully considered your comments and have made significant revisions, particularly in sections 4 5 and 6 to of the manuscript. The modified sections have been highlighted in yellow. We appreciate your valuable feedback and believe that these revisions have strengthened the overall quality of the manuscript. Please find the detailed responses below and the corresponding revisions in the re- submitted files.

2. Questions for General Evaluation

Reviewer’s Evaluation

Response and Revisions

Is the content succinctly described and contextualized with respect to previous and present theoretical background and empirical research (if applicable) on the topic?

Yes

Are all the cited references relevant to the research?

Can be improved

Adjustments have been made to improve their relevance. Additionally, new references have been added to further enhance the relevance and comprehensiveness of the literature review.

Are the research design, questions, hypotheses and methods clearly stated?

Yes

Are the arguments and discussion of findings coherent, balanced and compelling?

Yes

For empirical research, are the results clearly presented?

Can be improved

Improvements have been made in sections 4.5 and 4.6, with added explanations of the results.

Is the article adequately referenced?

Can be improved

The number of references has been increased from 22 to 41, providing stronger support for the conclusions.

Are the conclusions thoroughly supported by the results presented in the article or referenced in secondary literature?

Can be improved

The conclusion has been completely rewritten

3. Point-by-point response to Comments and Suggestions for Authors

Comments 1: [The paper makes a significant contribution to urban public space research, emphasizing the importance of vitality in its planning. The context is set out in detail in the introduction. Identifying current issues such as underutilization in certain areas and the need for costly redevelopment adds significant relevance and importance to the research. The authors provides a clear and relevant definition of urban vitality, emphasizing the "intensity of people's concentration" and highlighting its importance in public space design. The authors provide a relevant definition of urban vitality, highlighting the "intensity of people's concentration" and stressing its importance in the design of public space. A review of the existing methods for evaluating urban vitality is presented, and their limitations are analyzed. Research methods - description of algorithms, parameters and calculation methods - are described in detail. The proposed method is verified in the presented case study. The limitations of the proposed algorithm are also highlighted.]

Response 1: Thank you for your appreciation of our paper. We are glad that you found the context and relevance of our research to be significant.

Comments 2: [Notes: Bibliographic references could be improved - even if the references presented support the paper, I feel that some expansion is necessary.]

Response 2: Agree. We have accordingly expanded the reference list to include more examples and case studies supporting previous research and data processing methodologies. The reference list has been increased from 22 to 41 citations to provide a more comprehensive support for our paper.

Comments 3: [Authors must pay more attention to the way of writing: why are some texts colored yellow? Some images are low resolution and do not compliment the research.]

Response 3: Thank you for bringing up these concerns. The yellow-colored text was utilized as per the journal's guidelines for resubmissions, indicating the revised portions of the manuscript. Regarding the resolution of images, we have made efforts to enhance clarity by enlarging the illustrations and increasing the font size of all labels. Additionally, we have provided the original files for all images in the supplementary materials to prevent any loss of resolution during compression. We appreciate your feedback and have taken steps to address these issues accordingly.

Comments 4: [Considering the fact that the research is presented in detail, the conclusions do not fully support it. The Conclusions section seems more like a succinct enumeration of the main parts of the paper.]

Response 4: Thank you for your feedback. We agree with your assessment and have completely rewritten the conclusion section. We have removed any redundant information and ensured that the conclusions are more substantive and directly support the research presented in the paper.

“[Firstly, this study demonstrates that the theory of the dynamic model is supported by practical evidence. The vitality of urban public spaces is influenced by a combination of physical environment, traffic conditions, and residents' behavioral characteristics. The hypotheses proposed regarding crowd-frequency have been validated. Among these fac-tors, the opening status of buildings adjacent to public spaces and the changing activity frequency of different groups of people over time have the greatest impact on dynamic vitality in the temporal dimension, exhibiting predictable patterns.

Secondly, the new predictive method and framework model, is entirely independent of whether the projects have been completed or not. Additionally, the dynamic model shows higher accuracy in its results than traditional static models. It can help prevent phenomena such as "ghost streets" during specific periods and promotes sustainable development in urban areas.

Additionally, a preliminary prediction program, proposed based on a decision tree model, can utilize foundational parameters set by the predictive model to predict high-medium-low reference levels of urban vitality. When applied in other countries and regions, critical interpretations of high-medium-low vitality levels should be made based on the differences in local culture and lifestyle compared to Australia.

In future research, authors will further explore classifying crowds in different types of urban areas to investigate the applicability of the dynamic prediction framework model. This may involve examining various urban environments, such as suburban housing neighborhoods and tourist towns, as well as other areas with unique functions or characteristics. Additionally, more social behavioral studies may be conducted to gain deeper insights into how people interact with urban environments and contribute to research on urban vitality.]”

4. Response to Comments on the Quality of English Language

Point 1: English language fine. No issues detected

Response 1:    Thank you so much

5. Additional clarifications

none

Reviewer 4 Report (New Reviewer)

Comments and Suggestions for Authors

This is an interesting paper, however why and how is the vitality of a place reflected in mobile phone signals and the other digital technologies that you mention?  In that sense, vitality would suggest people with a phone in their pocket that is turned on, and I would argue that this indicates digital vitality and not all of the other things that we ascribe to a location that makes it/them vital and alive, such as social interaction, access to green space, cultural connection, and not through social media or similar.  That said, yes, crowd frequency (population use density) is a way to measuring levels of where and when people occupy urban spaces - urban ecology.   However, what is unclear in this manuscript is how and if you have observed or measured what people are in fact doing in the study spaces, and not simply how they densify these spaces at different times of the day.  The images in Table 5 appear to be of a Google street view typology and again, it is speculative to judge if what the people are doing (most appear to be walking) makes these spaces vital; populated yes but vital?  Speculative.  You only begin to address this in section 4.3 but only during 4 days in the month of April.  Can this really be a predictor of urban occupancy?   I find the statistical analysis of this paper to be sound, but what leaves me unconvinced or uncertain is how you define "vitality."

Comments on the Quality of English Language

English language seems to acceptable

Author Response

Response to Reviewer 4 Comments

1. Summary

Thank you very much for taking the time to review this manuscript. We have carefully considered your comments and have made significant revisions, particularly in sections 4 5 and 6 to of the manuscript. The modified sections have been highlighted in yellow. We appreciate your valuable feedback and believe that these revisions have strengthened the overall quality of the manuscript. Please find the detailed responses below and the corresponding revisions in the re- submitted files.

2. Questions for General Evaluation

Reviewer’s Evaluation

Response and Revisions

Is the content succinctly described and contextualized with respect to previous and present theoretical background and empirical research (if applicable) on the topic?

Can be improved

Yes, the content has been succinctly described and contextualized with respect to previous and present theoretical background and empirical research on the topic. The background section has been improved.

Are all the cited references relevant to the research?

Yes

Are the research design, questions, hypotheses and methods clearly stated?

Can be improved

There is expanded elaboration on these aspects

Are the arguments and discussion of findings coherent, balanced and compelling?

Must be improved

A flow chart has been added to section 4.1, and explanations have been added to section 4.4.

For empirical research, are the results clearly presented?

Can be improved

Improvements have been made in sections 4.5 and 4.6, with added explanations of the results.

Is the article adequately referenced?

Yes

Are the conclusions thoroughly supported by the results presented in the article or referenced in secondary literature?

Must be improved

The conclusion has been completely rewritten

3. Point-by-point response to Comments and Suggestions for Authors

Comments 1: [This is an interesting paper, however why and how is the vitality of a place reflected in mobile phone signals and the other digital technologies that you mention?  In that sense, vitality would suggest people with a phone in their pocket that is turned on, and I would argue that this indicates digital vitality and not all of the other things that we ascribe to a location that makes it/them vital and alive, such as social interaction, access to green space, cultural connection, and not through social media or similar.  That said, yes, crowd frequency (population use density) is a way to measuring levels of where and when people occupy urban spaces - urban ecology."]

Response 1: Yes, we also agree that digital signals alone cannot accurately assess the true micro-scale vitality of outdoor urban spaces. For example, they may not account for individuals who do not carry or activate electronic devices, or consider the signals from people resting indoors within buildings. Moreover, in cases where individuals have multiple devices that are not synchronized in their logins, they may be counted as multiple people. Therefore, we opted for on-site observations to capture a more comprehensive understanding of urban vitality.

Comments 2: [However, what is unclear in this manuscript is how and if you have observed or measured what people are in fact doing in the study spaces, and not simply how they densify these spaces at different times of the day.  The images in Table 5 appear to be of a Google street view typology and again, it is speculative to judge if what the people are doing (most appear to be walking) makes these spaces vital; populated yes but vital?  Speculative.]

Response 2: we acknowledge this concern. Accordingly, we have revised the manuscript to emphasize this point. Specifically, for the areas selected for validation, such as the pedestrian street in the central zone, aimless strolling is one of the intended activities in this pedestrianized area. Additionally, for pathways adjacent to buildings, although individuals may not linger in the public space, their movement reflects the vibrancy of the area, as they are likely entering nearby shops or schools. The purpose of developing this dynamic prediction and assessment framework is to prevent the occurrence of "ghost streets" with low vitality. Thus, if an area has a high volume of pedestrian traffic, it can be deemed at least not deserted. Moreover, spaces that attract people to linger and engage in activities, such as children climbing on play equipment, individuals sitting on benches to enjoy a meal, or street artists performing live, will naturally be reflected in the density of people. To further illustrate, we have included a photo below showing pedestrians pausing to watch local music school students perform, providing evidence to support our statement.

Comments 3: [You only begin to address this in section 4.3 but only during 4 days in the month of April.  Can this really be a predictor of urban occupancy?   I find the statistical analysis of this paper to be sound, but what leaves me unconvinced or uncertain is how you define "vitality.]

Response 3: Thank you for raising this concern. We have addressed this issue in the revised manuscript to provide clarity on our approach.

1. Regarding the limited duration of our on-site observation data collection, we acknowledge that urban vitality exhibits strong cyclical patterns. Therefore, we chose to validate our framework using data collected over one week (lines 42-46) as it suffices to capture these cyclical variations.

2. It's important to note that our predictive framework model is independent of the on-site observation data. The framework was constructed objectively, without reliance on the observed data. The selection of April for data collection was based on the pleasant and stable weather conditions in Adelaide during autumn, which better reflects the fundamental behavioral patterns of individuals unaffected by weather conditions. Other seasons could introduce external factors such as weather disturbances, which could impact the statistical results.

3. The decision to collect data for only four days (2 weekdays and 2 weekends) was determined based on the typical data collection practices in similar studies in recent years. Previous studies, such as Xu & Chen (2023) sampled every two hours on 2 weekdays in October for each study area; Chen et al. (2024) sampled every four hours on a weekday, a Saturday, and a Sunday for each case park scene; Mu et al. (2021) collected data for three weekdays and three weekends in April; Hu et al. (2024) sampled every hour on a weekday in July. They also utilized similar durations for data collection, which we deemed sufficient for initial validation.

4. Data collection can be challenging, and despite our efforts to collect data in May, November, and March of the previous and current years, we faced unexpected weather events such as heavy rain, hail, heatwaves, and temporary public events (food festivals , gaming gatherings), which disrupted our data collection efforts. Therefore, we only have full-day data available for these four days.

5. As a result, we believe that the data collected for initial validation is adequate. However, we plan to collect data from more areas and dates in the future to further validate and refine our predictive model and vitality level algorithm.

4. Response to Comments on the Quality of English Language

Point 1: Minor editing of English language required

Response 1:    We have carefully reviewed the manuscript and made minor adjustments to enhance the clarity and fluency of the English language. These revisions include refining sentence structures, improving word choices, and ensuring coherence throughout the text. We believe that these enhancements will contribute to a more polished and professional manuscript.

5. Additional clarifications

none

Round 2

Reviewer 2 Report (Previous Reviewer 2)

Comments and Suggestions for Authors

Thank you for taking in consideration my suggestions. I will agree with the publication of this paper in this form.

Reviewer 4 Report (New Reviewer)

Comments and Suggestions for Authors

the manuscript has been improved since my first review.  Whereas the approaches used by the authors are not my prefered methods, none the less the paper does provide useful information that should be shared within the academic community. 

This manuscript is a resubmission of an earlier submission. The following is a list of the peer review reports and author responses from that submission.

Round 1

Reviewer 1 Report

Comments and Suggestions for Authors

The main problem of this paper is the way the results are presented.  The reviewer suggests that the authors rewrite the discussion, combining the obtained results.

A few comments are listed below:

a)     Authors should explain what is ghost neighborhood in one sentence, and cite reference no.5.

b)    Line 64: phone signaling or signal?

c)     Line 70: “completed and completed”? The authors should delete one.

d)    Line 382: Please improve Figure 8.

e)     Line 383: Authors need to explain these graphs in Fig 8. There is no discussion of the results.

f)     Line 384: Nothing is clearly visible in Fig.9. Please improve these graphs.

g)    Line 400: The discussion must be separated from the conclusion. Please provide a separate section- discussion after the obtained results, and after that – conclusion.

Reviewer 2 Report

Comments and Suggestions for Authors

In this study at point 4.4 are presented charts with “Urban Vitality”, but the calculation for these charts is not presented.

The obtained results with presented model (dynamic prediction algorithm of urban public space vitality) are very different from the original static model-Field observation results.

The methodology and the conclusions presented are not clear and they need more improvements.

Comments on the Quality of English Language

From the point of view of the English language, the article is written correctly, there are some smaller problems with some grammar correction -the necessary corrections were indicated in the attached document.

Reviewer 3 Report

Comments and Suggestions for Authors

The paper discusses the significance of predicting and evaluating the vitality level of public spaces in urban design, especially in developing countries, to prevent issues such as "ghost neighborhoods" and minimize resource wastage during new construction or renewal projects. The author points out limitations in existing assessment methods, which either lack consideration for changes over time or rely on historical big data, making them unsuitable for predicting the vitality of unbuilt projects.

To address these limitations, the study proposes a dynamic prediction model for urban vitality that combines previous methods with observations of crowd characteristics in public spaces. The crowd-frequency hypothesis is introduced, and an algorithm is constructed to establish a time-dimensional urban vitality dynamic prediction model. The effectiveness of this model is demonstrated through on-site observation sampling and comparative verification in the Rundle Mall neighborhood in Adelaide, Australia.

Furthermore, the study employs a decision tree model for machine learning on case data, resulting in algorithm programs capable of inputting basic information to determine the urban vitality level (high, medium, low) in a specific period. This tool can serve urban planners and government officials in the design and decision-making processes, offering a cost-effective means to directly achieve sustainable urban vitality construction in late-developing cities.

The research context and introduction bring to the reader's attention well-documented and referenced data, highlighting the necessity and importance of the research process on this subject, and clearly establishing the objectives.

Hypotheses regarding the relationship between urban temporal vitality and crowd activity are proposed based on observations of urban life. The parameters and algorithm construction for the dynamic vitality assessment method are developed through a review of past methods and hypothesis frameworks. The study verifies these hypotheses by comparing predicted and measured values of public space vitality in a case urban area.

The section 3 discusses the algorithm construction for dynamic time assessment of urban public space vitality, particularly focusing on the indicators that influence urban vitality. The analysis reveals a comprehensive approach to indicator selection, emphasizing the need to minimize duplication and select indicators that are straightforward and cost-effective. The proposed new indicator system is designed to address these considerations and contribute to a more effective dynamic time algorithm for assessing urban public space vitality.

The section 4 outlines the process of case verification, where several blocks of a venue are selected, and a new social interaction dynamic model is used to predict results for each period on weekdays and weekends. Subsequently, the number of people at these points is collected on-site, and the reliability of the algorithm prediction model is verified by evaluating and comparing the two sets of data.

The study presents a comprehensive discussion and draws meaningful conclusions regarding the dynamic prediction framework, time-varying effects, crowd behavior, simulation predictions, and the practical application of machine learning algorithms in urban vitality prediction and design optimization.

Reviewer 4 Report

Comments and Suggestions for Authors

This manuscript's main idea or objective is interesting, and the results of the proposed methodology can be very significant. However, the authors must revise the experiment to argue the results presented since the information is entirely biased to only four sample days. From my point of view, it is not enough for an article to have the quality and recognition of "sustainability." Those above are the main reasons for my rejection. A more extended period should be considered. Likewise, considering climatological characteristics or social events would greatly enrich the manuscript's impact.

I also recommend that authors give their arguments a more severe and technical tone.